# Development of a Noninfectious Japanese Encephalitis Virus Replicon for Antiviral Drug Screening and Gene Function Studies

**DOI:** 10.3390/v17060759

**Published:** 2025-05-27

**Authors:** Yang Yang, Jiayang Zheng, Yafang Lin, Yan Zhang, Qianming Zhao, Hailong Zhang, Junjie Zhang, Zongjie Li, Ke Liu, Beibei Li, Donghua Shao, Yafeng Qiu, Zhiyong Ma, Jianchao Wei

**Affiliations:** 1Shanghai Institute of Infectious Disease and Biosecurity, Shanghai Veterinary Research Institute, Chinese Academy of Agricultural Sciences, Shanghai 200241, China; 15776623916@163.com (Y.Y.); jiayangzheng1997@163.com (J.Z.); 13065714060@163.com (Y.L.); zhanghailong1997@163.com (H.Z.); m17317271403@163.com (J.Z.); lizongjie@shvri.ac.cn (Z.L.); liuke@shvri.ac.cn (K.L.); lbb@shvri.ac.cn (B.L.); shaodonghua@shvri.ac.cn (D.S.); yafengq@shvri.ac.cn (Y.Q.); 2College of Veterinary Medicine, Nanjing Agricultural University, Nanjing 210095, China; 18672550235@163.com; 3College of Veterinary Medicine, Henan Agricultural University, Zhengzhou 450046, China; zhaoqm2022@163.com

**Keywords:** Japanese encephalitis virus, replicon, green fluorescent protein, antiviral drug screening, RNA replication

## Abstract

Viral replicons are efficient tools to understand the mechanisms of viral replication and screen antiviral drugs. In this study, a viral-cDNA-based replicon of Japanese encephalitis virus (JEV), which is the causative agent of Japanese encephalitis, was constructed by replacing the viral structural proteins with a green fluorescent protein (JEV-GFP replicon). The resulting JEV-GFP replicon was used as a tool to screen antiviral drugs targeting JEV nonstructural proteins, and the five compounds JNJ-A07, HZ-1157, NITD-2, quinine, and NITD008 were obtained, which significantly inhibited the replication of the JEV-GFP replicon and JEV in vitro, and the properties of these five compounds were also analyzed. The CC_50_, EC_50_, and SI indices of these five compounds were analyzed. In addition, the JEV-GFP replicon was used as a tool to identify the residues of viral nonstructural proteins involved in RNA replication, and the cysteine residue at position 4 of nonstructural protein 1 was found to be essential for JEV RNA replication. These data suggested that the noninfectious JEV-GFP replicon could be used as tool for different purposes, such as antiviral drug screening and gene function studies.

## 1. Introduction

Japanese encephalitis virus (JEV) is the causative agent of Japanese encephalitis (JE) [1], which is primarily endemic to the Asia–Pacific region, and more than 3 billion people are at risk [2]. It is recognized as a zoonotic virus that is transmitted among mammals, including humans and pigs, and wild birds via mosquito vectors [3]. JEV belongs to the genus *Orthoflavivirus* of the family *Flaviviridae*, which comprises more than 70 species, such as West Nile virus (WNV), dengue virus (DENV), and Zika virus (ZIKV).

JEV is an enveloped, single-stranded, positive-sense RNA virus with a genome of approximately 11 kb in length. The JEV genome consists of two untranslated regions (UTRs) at the 5′- and 3′-ends of the genome and one open reading frame encoding a single polyprotein [4]. The polyprotein is proteolytically processed by a combination of viral and host proteases after translation, generating three structural proteins (C, prM/M, and E) and seven nonstructural proteins (NS1, NS2A, NS2B, NS3, NS4A, NS4B, and NS5) [5]. The structural proteins play essential roles in virion assembly, while the nonstructural proteins are involved in viral RNA replication, virion assembly, and the escape from innate immune responses [6].

Although the burden of JE has been greatly reduced by vaccination [7], it still has a global incidence of approximately 67,900 cases per year [8]. Currently, no specific anti-JEV therapeutics are available; thus, it is important to develop effective and safe antiviral drugs for the worldwide treatment of JEV. The nonstructural proteins are primarily involved in RNA replication, virus assembly, and the modulation of host responses, and they have been considered as promising drug targets for the development of anti-JEV drugs [9,10].

The nonstructural proteins of flaviviruses play critical roles in viral RNA replication, while their structural proteins are dispensable for viral RNA replication [11]. Based on these findings, a variety of self-replicating sub-genomic replicons of flaviviruses, such as DENV, YFV, and WNV, which contain all the nonstructural proteins and large deletions of the structural proteins, have been generated [12,13] and used for different purposes, such as the screening of antiviral drugs targeting the nonstructural proteins. This is because these replicons can be used to examine viral RNA replication exclusively without the intervention of other events of flavivirus infection, such as viral entry into host cells, progeny virus assembly and release, and so on [14,15].

JEV replicons expressing *Renilla* luciferase and the enterovirus-71 epitope have been described and used for antiviral screening and developing viral-replicon-based vaccines [1,16]. In this study, a JEV-cDNA-based replicon expressing the green fluorescent protein (GFP), the JEV-GFP replicon, was constructed by replacing the viral structural proteins with GFP, and it was used as a tool to screen antiviral drugs targeting JEV nonstructural proteins and for the identification of the NS1 residue required for viral RNA replication.

## 2. Materials and Methods

### 2.1. Cells, Virus, and Antibodies

BHK-21, 293T, and A549 cells were maintained in a high-glucose Dulbecco Modified Eagle Medium (DMEM) (Invitrogen, Carlsbad, CA, USA) containing 10% fetal bovine serum (FBS) (Gibco, Shanghai, China) and 1% penicillin–streptomycin at 37 °C with 5% CO_2_. The JEV SD12 strain was used for the extraction of viral RNA. An EGFP-labeled recombinant JEV GI strain (rJEV-EGFP) was gifted by Dr. Chenxi Li of Yangzhou University. Antibodies used included a monoclonal antibody specific to JEV NS1 (GTX131370, GeneTex, Shanghai, China), an anti-GAPDH monoclonal antibody (ab9485, Abcam, Shanghai, China), HRP labeled goat anti-rabbit IgG antibodies, and goat anti-mouse IgG antibodies coupled with Alexa Fluor^®^488 (Thermo Fisher Scientific, Shanghai, China).

### 2.2. Construction of the JEV-GFP Replicon Expressing Green Fluorescent Protein (GFP)

Viral RNA was extracted from the JEV SD12 strain using the RNA Easy Fast Tissue/Cell Kit (DP451, TIANGEN, Beijing, China) and reverse-transcribed into cDNA using an Evo M-MLV RT premix (AG, Changsha, Hunan, China). The resulting cDNA template was used to amplify three gene fragments, including a fragment from 5′-UTR to the sequence encoding the first 34 amino acids at the N-terminus of the C protein, a fragment encoding amino acids from the first 22 amino acids at the C-terminus of the E protein to the first 320 amino acids at the N-terminus of the NS3 protein, and a fragment encoding amino acids from the first 320 amino acids at the N-terminus of the NS3 protein to the NS5 protein plus the 3′-UTR sequence, using a PCR with specific primers (Appendix A). The GFP gene fragment was also amplified with a PCR using specific primers (Appendix A). The amplified gene fragments were assembled using fusion PCR technology to obtain the full-length cDNA sequence of the JEV-GFP replicon. The full-length cDNA sequence of the JEV-GFP replicon was subsequently inserted into a low-copy POK vector containing the CMV promoter that was linearized by *BamH* I and *Xba* I double digestion to generate a plasmid harboring the viral cDNA-based JEV-GFP replicon. The resulting plasmids were confirmed using DNA sequencing (Huajin biological company, Shanghai, China). The plasmids containing gene mutations and/or deletions were discarded. To construct the JEV-GFP mutant replicon (JEV-GFP-NS1/C4A) with a cysteine-to-alanine mutation at position 4 of NS1 (NS1/C4A), a fragment containing an NS1/C4A point mutation was amplified using PCR with the point mutation primers (Appendix A) and using the plasmid JEV-GFP replicon as a template. The resulting fragment with an NS1 C4A point mutation was inserted into the plasmid JEV-GFP replicon and digested using *Daz* I and *Sma* to generate a plasmid of the JEV-GFP-NS1/C4A replicon. The resulting plasmids were confirmed using DNA sequencing.

### 2.3. Transfection of the JEV-GFP Replicon

BHK-21 cells were transfected with a plasmid harboring the JEV-GFP replicon using Lipofectamine™ 3000 (L3000015, Invitrogen, Carlsbad, CA, USA) according to the manufacturer’s instructions and incubated in DMEM containing 2% FBS and 1% penicillin–streptomycin at 37 °C with 5% CO_2_. The replication of the JEV-GFP replicon was determined using qRT-PCR and a Western blot analysis. The fluorescence of GFP was visualized under a fluorescence microscope, and fluorescence intensities were measured using a multi-detector microplate reader.

### 2.4. Western Blot, Immunofluorescence (IFA), and qRT-PCR Analyses

Western blot, IFA, and qRT-PCR analyses were performed as described in a previous study [17]. The primers targeting viral NS1 and NS3 (Appendix A) were designed based on the gene sequence of the JEV SD12 strain.

### 2.5. Antiviral Drug Screening

The library of bioactive compounds used in this study (Appendix A) was purchased from the MCE company (Shanghai, China). BHK-21 cells were transfected with a plasmid harboring the JEV-GFP replicon and incubated for 6 h. The culture media were replaced with fresh culture media containing a bioactive compound at the indicated concentration (Appendix A), and the cells were incubated for 48 h. The fluorescence of GFP was visualized under a fluorescence microscope, and fluorescence intensities were measured and calculated using a multi-detector microplate reader. The expression of viral NS1 was detected using qRT-PCR.

### 2.6. Cytotoxicity Assay

JNJ-A07, HZ-1157, NITD-2, quinine, and NITD008 (purity ≥ 98.0%) were dissolved in DMSO at a concentration of 10 mM and stored at −80 °C in the dark. Cell viability was measured using the Cell Counting Kit-8 (CCK-8), which measures the cellular dehydrogenase activity (Beyotime, Shanghai, China), according to the manufacturer’s protocol. Briefly, A549 cells were cultured in DMEM containing 10% FBS and 1% penicillin–streptomycin for 24 h in a 96-well plate. The medium was then replaced with DMEM containing 10% FBS, and serial dilutions of JNJ-A07, HZ-1157, NITD-2, quinine, and NITD008 (0.00192, 0.096, 0.048, 0.24, 1.2, 6, 30, and 150 μM) were performed and incubated for 48 h. To measure cell viability, the CCK-8 solution (10 μL per well) was added to each well and incubated at 37 °C for 2 h in the dark. The absorbance was measured at 450 nm using a multi-detector microplate reader. The 50% cytotoxic concentration (CC_50_) value was defined as the concentration of JNJ-A07, HZ-1157, NITD-2, quinine, and NITD008 that reduced the absorbance of treated cells by 50% when compared to control wells that were mock-treated with DMSO. The CC_50_ value was calculated using GraphPad Prism 8.0 (GraphPad software, La Jolla, CA, USA).

### 2.7. Analysis of the Inhibitory Effect of Antiviral Drugs

For the analysis of the inhibition of JEV-GFP replicon replication, BHK-21 cells were transfected with a plasmid harboring the JEV-GFP replicon and incubated for 6 h. The medium was then replaced with DMEM containing 2% FBS and 10 μM JNJ-A07, HZ-1157, NITD-2, quinine, and NITD008, and the cells were incubated for 48 h. The fluorescence of GFP was visualized under a fluorescence microscope, and fluorescence intensities were measured and calculated using a multi-detector microplate reader. The expression of viral NS1 was detected using qRT-PCR.

For the analysis of the inhibition of rJEV-EGFP replication, BHK-21, 293T, and A549 cells were infected with rJEV-EGFP at a multiplicity of infection (MOI) of 0.1 and incubated for 2 h. The medium was then replaced with DMEM containing 2% FBS and 10 μM JNJ-A07, HZ-1157, NITD-2, quinine, and NITD008, and the cells were incubated for 48 h. The fluorescence of GFP was visualized under a fluorescence microscope, and fluorescence intensities were measured and calculated using a multi-detector microplate reader. To determine the 50% effective concentration (EC_50_), A549 cells infected with rJEV-EGFP were treated with different concentrations of JNJ-A07 (0.0000256, 0.000128, 0.00064, 0.0032, 0.016, 0.08, 0.4, 2, and 10 μM) and HZ-1157, NITD-2, quinine, and NITD008 (0.0000512, 0.0002560, 0.00128, 0.0064, 0.032, 0.16, 0.8, 4, 20). The medium was then replaced with DMEM containing 2% FBS and 10 μM JNJ-A07, HZ-1157, NITD-2, quinine, and NITD008, and the cells were incubated for 48 h. The expression of the viral NS1 gene in each group was detected using qRT-PCR. The inhibition rate was calculated according to the following formula: viral inhibition rate = (1 − viral NS1 gene expression in drug-treated group/viral NS1 gene expression in control DMSO-treated group) × 100% [18]. The EC_50_ was defined as the concentration required to reduce NS1 gene expression by 50% when compared to control wells mock-treated with DMSO and was calculated using GraphPad Prism 8.0 [19]. The Selectivity Index (SI) is used as a safety indicator to assess the efficacy of a drug, where SI = CC_50_/EC_50_ [20]. When SI > 1, it indicates that the drug is safe, and the higher the SI value, the better the drug’s safety and efficacy.

### 2.8. Statistical Analysis

Statistical significance was evaluated using two-tailed unpaired Student’s *t*-tests; *p* < 0.05 was considered to be statistically significant.

## 3. Results

### 3.1. Construction of the JEV-GFP Replicon

The JEV-GFP replicon expressing the GFP was constructed by replacing most of the viral structural proteins (C, prM, and E) with the GFP using the full-length cDNA of the JEV SD12 strain as the backbone (Figure 1A). The first 34 amino acids of the C protein (C_34_) and the last 22 amino acids of the E protein (E_22_) were retained in the replicon to ensure the correct replication of the replicon. The fragment of the JEV-GFP replicon containing GFP, all nonstructural protein genes, and the 5′- and 3′-UTRs were obtained using fusion PCR technology and subsequently ligated to a low-copy POK vector containing a CMV promoter to generate a plasmid of the JEV-GFP replicon (Figure 1A).

The resulting JEV-GFP replicon was transfected into BHK-21 cells, and the replication of the JEV-GFP replicon in the transfectants was determined using qRT-PCR. The levels of the viral NS1 gene increased, and they peaked at 48 h post-transfection (Figure 1B). The green fluorescence of the transfectants was also observed at 48 h post-transfection, when compared with that of the mock-transfected cells (Figure 1C). The transfectants were subsequently harvested for the detection of viral NS1 expression. The expression of NS1 in the transfectants was detected using a Western blot analysis with anti-NS1 antibodies (Figure 1D). Furthermore, the transfectants were stained with IFA and anti-NS1 antibodies, and the presence of NS1 was observed in the cells (Figure 1E). These results suggested that the JEV-GFP replicon was successfully constructed.

**Figure 1 viruses-17-00759-f001:**
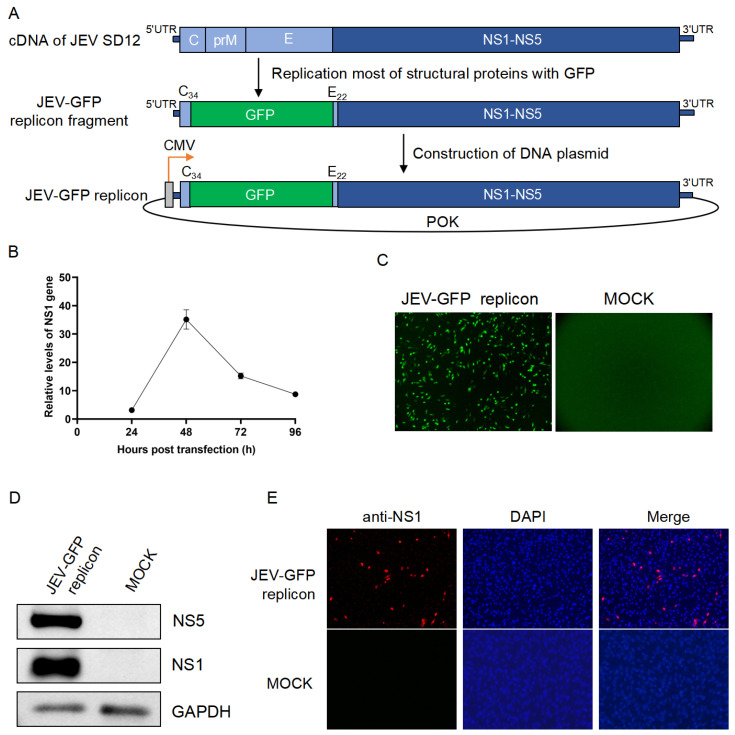
**Construction of JEV-GFP replicon.** (**A**) Schematic representation of construction of JEV-GFP replicon. (**B**) BHK-21 cells were transfected with JEV-GFP replicon and levels of NS1 gene were determined using qRT-PCR. Relative levels of NS1 gene expression in JEV-GFP-replicon-transfected cells were normalized to level of internal control, GAPDH. (**C**) Green fluorescence visualized under fluorescence microscope at 48 h post-transfection in BHK-21 cells transfected with JEV-GFP replicon. (**D**) Detection of NS1 and NS5 expression using Western blot analysis with anti-NS1 antibodies and anti-NS5 antibodies. (**E**) Detection of NS1 expression (red) using IFA with anti-NS1 antibodies. Nuclei were stained with DAPI (blue). Scale bars, 200 µm.

### 3.2. The JEV-GFP Replicon Used for Antiviral Drug Screening

Virus replicons have been used as tools for the screening of antiviral drugs [3]. Therefore, the JEV-GFP replicon was tested for its potential application in anti-JEV drug screening using a library of bioactive compounds that inhibit flavivirus replication. The BHK21 cells were transfected with the JEV-GFP replicon and subsequently treated with the bioactive compounds selected from the library of bioactive compounds. Among the 20 bioactive compounds that were tested, 16 reduced the green fluorescence intensities of the JEV-GFP replicon (Figure 2A, Appendix A) and 12 inhibited the NS1 gene expression of the JEV-GFP replicon (Figure 2B), with the compounds JNJ-A07, HZ-1157, NITD-2, quinine, and NITD008 showing the higher inhibitory effect.

The cytotoxicity of JNJ-A07 was determined by incubating A549 cells with JNJ-A07, HZ-1157, NITD-2, quinine, and NITD008 at different concentrations. JNJ-A07, HZ-1157, NITD-2, quinine, and NITD008 50% cytotoxic concentration (CC_50_) values were estimated at 29.86 μM, 21.22 μM, 49.17 μM, 45.76 μM, and 37.76 μM (Figure 3A). The cytotoxicities of JNJ-A07, HZ-1157, NITD-2, quinine, and NITD008 at 10 μM, which was used for determining the inhibition of JEV-GFP replicon replication, were further tested in BHK-21, 293T, and A549 cells, and no significant cytotoxicities were observed (Figure 3B). Therefore, 10 μM concentrations of JNJ-A07, HZ-1157, NITD-2, quinine, and NITD008 were used for the following experiments.

To confirm the inhibitory effect of JNJ-A07, HZ-1157, NITD-2, quinine, and NITD008 on JEV-GFP replicon replication, BHK-21 cells transfected with the JEV-GFP replicon were again treated with JNJ-A07 at a dose of 10 μM, and the replication of the JEV-GFP replicon was analyzed accordingly. The treatment of the transfectants significantly reduced the intensity of green fluorescence, when compared with those of the transfectants mock-treated with DMSO (Figure 3C,D). The inhibitory effects of JNJ-A07, HZ-1157, NITD-2, quinine, and NITD008 on JEV-GFP replicon replication were further determined using qRT-PCR. The levels of the NS1 gene (Figure 3E) were reduced in the transfectants treated with JNJ-A07, HZ-1157, NITD-2, quinine, and NITD008, when compared with those in the transfectants mock-treated with DMSO, suggesting that JNJ-A07, HZ-1157, NITD-2, quinine, and NITD008 are anti-JEV drug candidates. These data indicated that JEV-GFP replicons could be used as tools to preliminarily mass-screen anti-JEV drugs by measuring the green fluorescence intensity.

**Figure 2 viruses-17-00759-f002:**
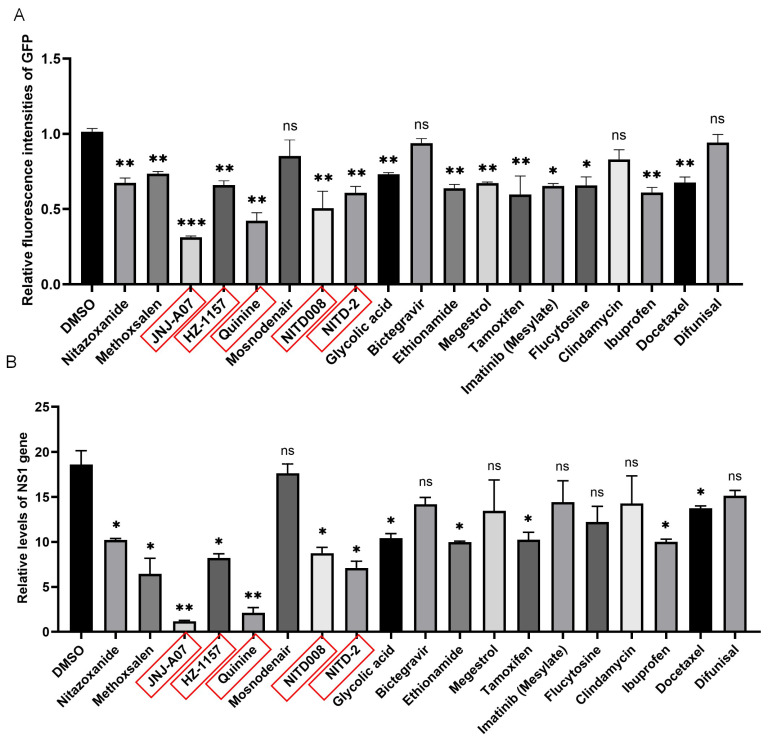
**JEV-GFP replicon used for antiviral drug screening.** BHK-21 cells were transfected with JEV-GFP replicon and subsequently treated with indicated bioactive compounds. (**A**) Fluorescence intensities were measured using multi-detector microplate reader. Relative fluorescence intensities in JEV-GFP-replicon-transfected and drug-treated cells were normalized to level of cells without transfection of JEV-GFP replicon and drug treatment and are presented relative to the level in JEV-GFP-replicon-transfected and DMSO-treated cells (set as 1). (**B**) Levels of NS1 gene expression were determined using qRT-PCR. Relative levels of NS1 gene expression in JEV-GFP-replicon-transfected and drug-treated cells were normalized to level of internal control, GAPDH. The drug-treated cells were compared with DMSO-treated control cells, and significant difference was determined using Student’s *t*-test. *, *p* < 0.05. **, *p* < 0.01. ***, *p* < 0.001. ns, no significant difference.

**Figure 3 viruses-17-00759-f003:**
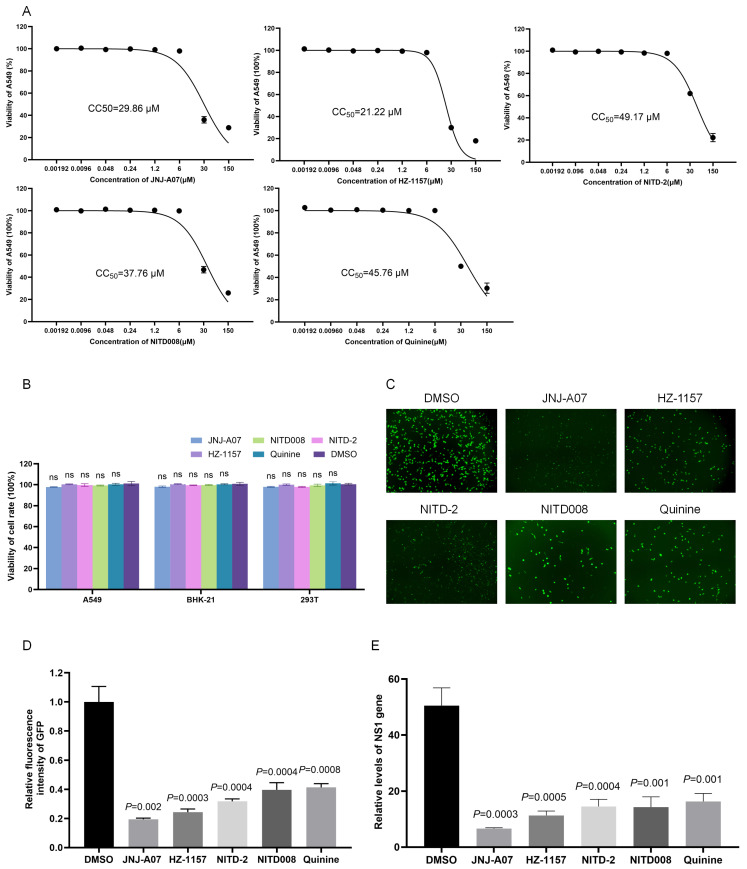
**Analysis of inhibitory effect of JNJ-A07 on replication of JEV replicon.** (**A**) A549 cells were treated with JNJ-A07, HZ-1157, NITD-2, quinine, and NITD008 at different concentrations (0.00192, 0.096, 0.048, 0.24, 1.2, 6, 30, and 150 μM) for 48 h. Viabilities of JNJ-A07-, HZ-1157-, NITD-2-, quinine-, and NITD008-treated cells were measured using Cell Counting Kit-8 and plotted. The 50% cytotoxic concentration (CC_50_) was analyzed using GraphPad Prism 8.0. (**B**) BHK-21, 293T, and A549 cells were treated with 10 μM JNJ-A07, HZ-1157, NITD-2, quinine, and NITD008 for 48 h, and viabilities of JNJ-A07-, HZ-1157-, NITD-2-, quinine-, and NITD008-treated cells were measured using Cell Counting Kit-8. (**C**–**E**) BHK-21 cells were transfected with JEV-GFP replicon and subsequently treated with 10 μM JNJ-A07, HZ-1157, NITD-2, quinine, and NITD008 for 48 h. Green fluorescence was visualized under fluorescence microscope (**C**). Fluorescence intensities were measured and calculated; scale bars, 200 µm (**D**). The levels of NS1 gene were determined (**E**). These drug-treated samples were compared with DMSO-treated control, and significant difference was determined using Student’s *t*-test. ns, no significant difference.

### 3.3. Antiviral Effect of JNJ-A07, HZ-1157, NITD-2, Quinine, and NITD008 on JEV Replication In Vitro

The anti-JEV effects of JNJ-A07, HZ-1157, NITD-2, quinine, and NITD008 were tested using an EGFP-labeled recombinant JEV GI strain (rJEV-EGFP). Three types of cells including BHK-21, 293T, and A549 cells were infected with rJEV-EGFP and subsequently treated with the JNJ-A07, HZ-1157, NITD-2, quinine, and NITD008 at a dose of 10 μM. The green fluorescence produced by rJEV-EGFP replication was determined at 48 h post-treatment. In response to the treatment, the intensities of green fluorescence were remarkably reduced in the cells treated with JNJ-A07, HZ-1157, NITD-2, quinine, and NITD008, when compared with those of the cells mock-treated with DMSO (Figure 4A,B). These data indicated that JNJ-A07, HZ-1157, NITD-2, quinine, and NITD008 had an inhibitory effect on JEV replication.

The inhibitory effects of JNJ-A07, HZ-1157, NITD-2, quinine, and NITD008 on JEV replication were further analyzed by treating JEV-infected cells with JNJ-A07, HZ-1157, NITD-2, quinine, and NITD008 at different concentrations. JEV replication was determined using qRT-PCR at 48 h post-treatment. The levels of the JEV NS1 gene were significantly decreased in the cells treated with JNJ-A07, HZ-1157, NITD-2, quinine, and NITD008 in a dose-dependent manner (Figure 4C). The 50% effective concentration (EC_50_) values of JNJ-A07, HZ-1157, NITD-2, quinine, and NITD008 were approximately 0.013 μM, 1.06 μM, 0.141 μM, 0.162 μM, and 0.113 μM (Figure 4D). Overall, these data confirmed the antiviral effect of JNJ-A07, HZ-1157, NITD-2, quinine, and NITD008 screened by the GFP-JEV replicon. The SI values of JNJ-A07, HZ-1157, NITD-2, quinine, and NITD008 were calculated and were are all relatively high, being 2296.92, 20.02, 348.72, 334.16, and 282.47, respectively (Table 1). The higher the SI value, the safer the drug. JNJ-A07 had the highest SI value, making it the safest and most effective of these five drugs.

### 3.4. The JEV-GFP Replicon Used for the Identification of the NS1 Residue Required for Viral RNA Replication

Flavivirus replicons have been used as tools to understand the role of viral nonstructural proteins [15,21]. Flavivirus NS1 co-localizes with double-stranded RNA and interacts with NS4B to modulate the early events of viral RNA replication [22]. The cysteine residue at position 4 (C4) of NS1, which is involved in the formation of disulfide bonds, plays a role in the stable folding phase of proteins [22,23] and may be required for JEV RNA replication. To test this hypothesis, the cysteine residue at position 4 of NS1 was replaced by alanine (A) to generate a JEV-GFP mutant replicon (JEV-GFP-NS1/C4A) with a cysteine-to-alanine mutation at position 4 (C4A) based on the JEV-GFP replicon (Figure 5A). The transfection of the JEV-GFP-NS1/C4A replicon and its parental JEV-GFP replicon expressing wild-type NS1 into BHK-21 cells showed that the green fluorescence was observed in the cells transfected with the JEV-GFP replicon but not in the cells transfected with the JEV-GFP-NS1/C4A replicon (Figure 5B). The fluorescence intensities in the cells transfected with the JEV-GFP-NS1/C4A replicon were measured at a basal level that was significantly lower than that in the cells transfected with the JEV-GFP replicon (Figure 5C). The analysis of NS3 gene expression using qRT-PCR indicated that the level of the NS3 gene in the cells transfected with the JEV-GFP-NS1/C4A replicon was significantly lower than that in the cells transfected with the JEV-GFP replicon (Figure 5D). The expression of the NS5 protein was detected in the cells transfected with the JEV-GFP replicon but not in the cells transfected with the JEV-GFP-NS1/C4A replicon (Figure 5E). These data indicated that the replication of the JEV-GFP-NS1/C4A replicon was impaired by the C4A mutation of NS1.

To further confirm this result, the JEV-GFP-NS1/C4A replicon was co-transfected with a plasmid expressing wild-type NS1 (NS1/WT) to complement wild-type NS1 in trans. A number of green fluorescent foci were visualized in the cells co-transfected with the JEV-GFP-NS1/C4A replicon and NS1/WT (JEV-GFP-NS1/C4A replicon + NS1/WT), when compared with the cells transfected with the JEV-GFP-NS1/C4A replicon alone (JEV-GFP-NS1/C4A replicon + vector) (Figure 5F). The analysis of fluorescence intensities and NS3 expression in the transfectants showed that the levels of fluorescence intensities (Figure 5G) and NS3 gene expression (Figure 5H) in the cells co-transfected with the JEV-GFP-NS1/C4A replicon and NS1/WT were remarkably higher than the basal levels measured in the cells transfected with the JEV-GFP-NS1/C4A replicon alone. This suggests that the function of NS1 was, at least in part, complemented in trans by the ectopic expression of wild-type NS1, further confirming that the replication of the JEV-GFP-NS1/C4A replicon was impaired by the C4A mutation of NS1. Overall, these data indicate that the JEV-GFP replicons could be used as tools to understand the role of viral nonstructural proteins in viral RNA replication and that the fourth cysteine residue site of NS1 is required for viral RNA replication.

## 4. Discussion

Replicons of flaviviruses, such as dengue virus [24], Zika virus [25], West Nile virus [26], etc., are used as tools to understand the roles of viral proteins and the mechanisms of viral replication as well as to screen antiviral drugs and develop viral-replicon-based vaccines. In this study, the JEV-GFP replicon expressing GFP, a JEV-cDNA-based replicator, was generated and used to screen anti-JEV drugs and identify the residue in viral NS1 required for JEV RNA replication.

The sequence corresponding to most structural proteins (C, prM, and E) was replaced with the GFP to construct the JEV-GFP replicon. The first 23 amino acids at the N-terminus of the viral C protein contain the flavivirus conserved cyclization sequence required for viral RNA replication and play an important role as cis-acting elements in regulating negative-sense RNA synthesis (also known as cycle-containing sequences) [12,21]. The last 24–30 amino acids at the C-terminus of the E protein contain the signaling sequence of the downstream NS1 protein and are responsible for the correct processing and translocation of NS1 and the correct topology of the nonstructural polyproteins on the endoplasmic reticulum [21,27]. Therefore, the first 34 amino acids of the C protein (C_34_) and the last 22 amino acids of the E protein (E_22_) were retained in the replicon to ensure the correct replication of the replicon. The analysis of viral NS1 expression and GFP fluorescence indicated that the constructed JEV-GFP replicon can self-replicate and express the nonstructural proteins and the GFP protein.

The flavivirus replicons are initiated by transfecting susceptible cells with the RNA replicon transcribed in vitro or introduced from a plasmid harboring a replicator cDNA under a cytomegalovirus (CMV) or SV40 promoter [28,29]. The JEV-GFP replicon constructed in this study was a JEV-cDNA-based replicator with a CMV promoter. The amplification of the JEV-cDNA-based replicon using a CMV promoter would be more efficient in the cytoplasm. In addition, the JEV-cDNA-based replicon would be more convenient to operate and more stable in the plasmid form because host cells are more likely to ingest a DNA-based replicon compared to an RNA-based replicon [17,27].

The JEV-GFP replicon harbored all nonstructural proteins (NS1, NS2A, NS2B, NS3, NS4A, NS4B, and NS5) that are promising drug targets for devising novel drugs against JEV because the nonstructural proteins are primarily involved in RNA replication, virus assembly, and the modulation of host responses [10,30]. Therefore, the JEV-GFP replicon was used to screen anti-JEV drugs using a library of bioactive compounds that inhibit flavivirus replication. Among the 20 tested bioactive compounds, the compounds JNJ-A07, HZ-1157, NITD-2, quinine, and NITD008 showed a higher inhibition of JEV-GFP replicon replication. The inhibitory effects of JNJ-A07, HZ-1157, NITD-2, quinine, and NITD008 were further confirmed in vitro using an EGFP-labeled recombinant JEV (rJEV-EGFP) in three types of cells and it was found that JNJ-A07, HZ-1157, NITD-2, quinine, and NITD008 significantly inhibited JEV replication in vitro. JNJ-A07, HZ-1157, NITD-2, quinine, and NITD008 50% had estimated cytotoxic concentration (CC_50_) values of 29.86 μM, 21.22 μM, 49.17 μM, 45.76 μM, and 37.76 μM, respectively. The EC_50_ values of JNJ-A07, HZ-1157, NITD-2, quinine, and NITD008 for inhibiting JEV replication were approximately 0.013 μM, 1.06 μM, 0.141 μM, 0.162 μM, and 0.113 μM, respectively. JNJ-A07 is an efficient inhibitor against the NS3–NS4B interaction of dengue virus (DENV) [30,31]. HZ-1157 is an inhibitor of the NS3/4A protease of hepatitis C virus (HCV) [32]. NITD-2 is a dengue virus (DENV) polymerase inhibitor [33]. NITD008 is a potent and selective flavivirus inhibitor [34]. Quinine is an alkaloid extracted from the bark of cinchona tree, which significantly inhibits DENV replication by reducing the synthesis of DENV RNA and viral proteins [35]. The five drugs preliminarily screened in this study—JNJ-A07, HZ-1157, NITD-2, quinine, and NITD008—do exhibit inhibitory effects on JEV replication, but the specific inhibitory mechanisms remain unclear. This can serve as a basis for future studies to explore the detailed mechanisms of JEV inhibition. These data suggested that the JEV-GFP replicon could be used as a tool to screen antiviral drugs and that JNJ-A07, HZ-1157, NITD-2, quinine, and NITD008 could be anti-JEV candidates. Moreover, the SI value of JNJ-A07 is the highest, and it is the safest and most effective. At present, there is no effective antiviral drug for JEV. Currently, the research drugs are mainly broad-spectrum antiviral drugs targeting nucleic acids, and drugs targeting viral replication via protease. The compounds screened in this study can all exert inhibitory effects on different stages of flavivirus replication. As JEV is also a flavivirus, these compounds may have a similar mechanism of action against JEV. This can be used as the basis for future research on the specific inhibitory mechanism of these drugs against JEV.

The JEV replicon expressing *Renilla* luciferase has been previously described and used for antiviral screening [36]. This replicon can be used in the high-throughput screening of antiviral drugs, but a commercial detection kit is required for measuring the luciferase activity. The JEV-GFP replicon constructed in this study can also be used to perform the high-throughput screening of antiviral drugs by measuring fluorescence intensities using a multi-detector microplate reader that can automatically capture and calculate fluorescent focus units without an additional detection kit, providing an alternative method for the high-throughput screening of anti-JEV drugs.

The nonstructural proteins are primarily involved in viral RNA replication [37], and viral NS1 modulates the early events of viral RNA replication [38]. Therefore, we selected the fourth cysteine residue site of NS1 that is involved in the formation of disulfide bonds and plays a role in the stable folding phase of proteins to examine whether the fourth cysteine residue site of NS1 was required for JEV RNA replication using the JEV-GFP replicon as a tool [22,39]. The cysteine-to-alanine mutation of the fourth cysteine residue site of NS1 (C4A) significantly reduced the viral NS3 gene and NS5 protein expression levels as well as the fluorescence intensities of GFP in the cells transfected with the JEV-GFP-NS1/C4A replicon, when compared with the results of cells transfected with the JEV-GFP replicon. In trans complementation of wild-type NS1 attenuated the reduced levels of viral NS3 gene expression and GFP fluorescence intensities caused by NS1 C4A mutation. These data indicated that the JEV-GFP replicon could be used as a tool to identify the residues of viral nonstructural proteins required for JEV RNA replication as well as to understand the role of viral nonstructural proteins in JEV RNA replication.

In conclusion, the JEV-GFP replicon expressing GFP, a JEV-cDNA-based replicator, was constructed by replacing the viral structural proteins with GFP. Subsequently, the JEV-GFP replicon was used as a tool to screen antiviral drugs targeting JEV nonstructural proteins from a library of bioactive compounds, and the compounds JNJ-A07, HZ-1157, NITD-2, quinine, and NITD008 were obtained, which significantly inhibited the replication of the JEV-GFP replicon and JEV in vitro, with EC_50_ values of 0.013 μM, 1.06 μM, 0.141 μM, 0.162 μM, and 0.113 μM, respectively. In addition, the JEV-GFP replicon was used as a tool to identify the residues of viral nonstructural proteins required for JEV RNA replication, and the fourth cysteine residue site of NS1 was found to be essential for JEV RNA replication. These data suggest that the JEV-GFP replicon could be used as a tool for different purposes, such as antiviral drug screening, understanding viral replication, and developing viral-replicon-based vaccines by replacing the GFP with viral antigens.

## Figures and Tables

**Figure 4 viruses-17-00759-f004:**
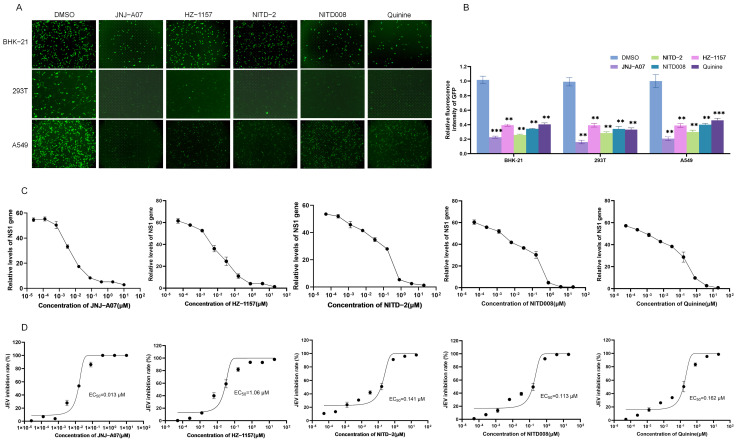
**Analysis of inhibitory effect of JNJ-A07 on JEV replication in vitro.** (**A**–**C**) BHK-21, 293T, and A549 cells were infected with rJEV-EGFP and subsequently treated with JNJ-A07, HZ-1157, NITD-2, quinine, and NITD008 at a dose of 10 μM for 48 h. The green fluorescence produced by rJEV-EGFP replication was visualized under fluorescence microscope; scale bars, 200 µm (**A**). Relative fluorescence intensities in rJEV-EGFP-infected and drug-treated cells were normalized to the level of cells without rJEV-EGFP infection and drug treatment, and are presented relative to the level in rJEV-EGFP-infected and DMSO-treated cells (set as 1) (**B**). (**C**,**D**) A549 cells were infected with rJEV-EGFP and subsequently treated with JNJ-A07 at different concentrations (0.0000256, 0.000128, 0.00064, 0.0032, 0.016, 0.08, 0.4, 2, and 10 μM), HZ-1157, NITD-2, quinine, and NITD008 at different concentrations (0.0000512, 0.0002560, 0.00128, 0.0064, 0.032, 0.16, 0.8, 4, 20) for 48 h. Levels of NS1 gene expression were determined using qRT-PCR (**C**). Inhibitory rates were calculated and plotted (**D**). JNJ-A07-, HZ-1157-, NITD-2-, quinine,- and NITD008-treated samples were compared with DMSO-treated control, and significant difference was determined using Student’s *t*-test. **, *p* < 0.01. ***, *p* < 0.001.

**Figure 5 viruses-17-00759-f005:**
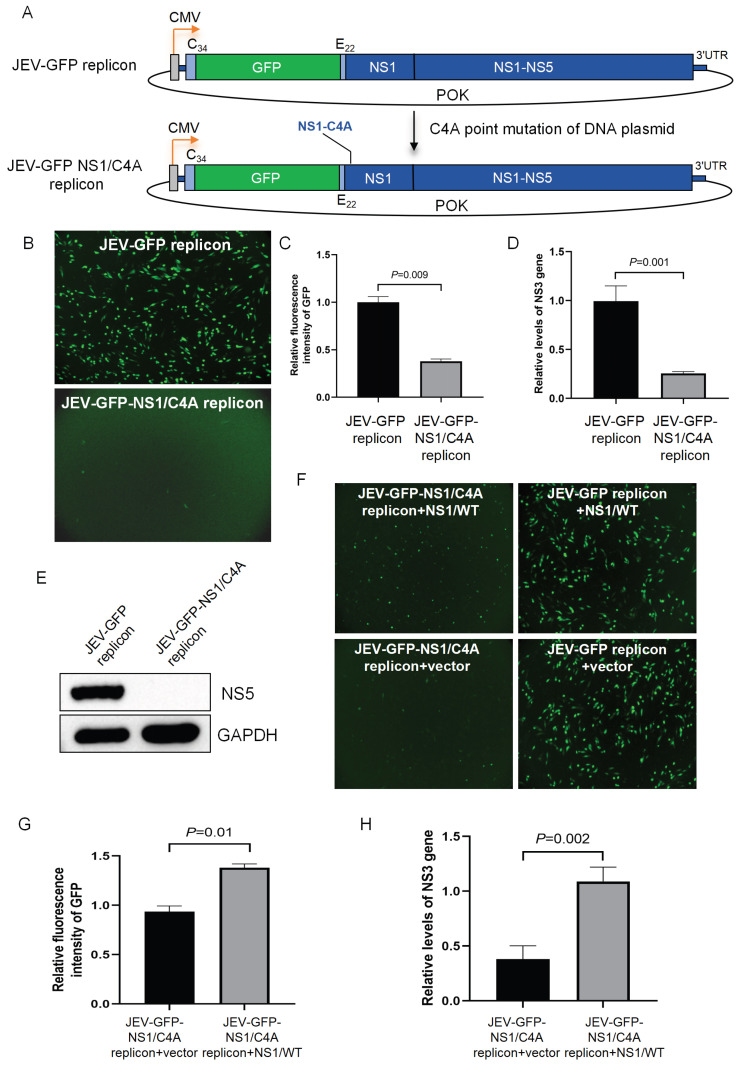
**Identification of NS1 residue required for viral RNA replication.** (**A**) Schematic representation of construction of JEV-GFP-NS1/C4A replicon. (**B**–**E**) BHK-21 cells were transfected with JEV-GFP-NS1/C4A replicon and its parental JEV-GFP replicon and incubated for 48 h. The green fluorescence was visualized under fluorescence microscope; scale bars, 200 µm (**B**), and the green fluorescence intensity was measured and calculated. Relative fluorescence intensities in replicon-transfected cells were normalized to level of mock-transfected cells and are presented relative to level in JEV-GFP-replicon-transfected cells (set as 1) (**C**). NS3 gene expression was examined using qRT-PCR. Relative levels of NS3 gene expression in replicon-transfected cells were normalized to level of internal control, GAPDH, and are presented relative to level in JEV-GFP-replicon-transfected cells (set as 1) (**D**). Expression of NS5 protein was detected using Western blot analysis (**E**). (**F**–**H**) BHK-21 cells were transfected with combinations of replicons and plasmid expressing wild-type NS1 (NS1/WT) (JEV-GFP-NS1/C4A replicon + NS1/WT, JEV-GFP-NS1/C4A replicon + vector, JEV-GFP replicon + NS1/WT, JEV-GFP replicon + vector). The green fluorescence was visualized under fluorescence microscope; scale bars, 200 µm (**F**). The green fluorescence intensities were measured and calculated. Relative fluorescence intensities in the transfectants were normalized to level of mock-transfected cells and are presented relative to level in cells co-transfected with JEV-GFP-NS1/C4A replicon and NS1/WT (set as 1) (**G**). Levels of NS3 gene were examined using qRT-PCR. Relative levels of NS3 gene expression in transfectants were normalized to level of internal control, GAPDH, and are presented relative to level in cells co-transfected with JEV-GFP-NS1/C4A replicon and NS1/WT (set as 1) (**H**). Significant difference between groups was determined using Student’s *t*-test.

**Table 1 viruses-17-00759-t001:** Test of reference drugs in anti-JEV assays.

	CC_50_	EC_50_	SI
JNJ-A07	29.86 µM	0.013 µM	2296.92
HZ-1157	21.22 µM	1.06 µM	20.02
NITD-2	49.17 µM	0.141 µM	348.72
NITD008	37.76 µM	0.113 µM	334.16
Quinine	45.76 µM	0.162 µM	282.47

## Data Availability

Data are contained within the article and Appendix A.

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
