# Peer review of "Development of a Noninfectious Japanese Encephalitis Virus Replicon for Antiviral Drug Screening and Gene Function Studies"

_viruses, 2025, doi:10.3390/v17060759_

Round 1

Reviewer 1 Report

Comments and Suggestions for Authors

This article presents the results of antiviral drug screening targeting the non-structural proteins of Japanese Encephalitis Virus (JEV) using a non-infectious JEV-GFP replicon; however, the information provided is too brief. While antiviral drugs that affect the expression of the JEV NS1 gene have been thoroughly screened, there has been no further investigation into the expression of other non-structural protein genes. Additionally, the mechanisms by which these antiviral drugs influence JEV NS1 gene expression have not been explored in depth, which is quite unfortunate. Furthermore, there are several questions that the author needs to address, such as:

1. Regarding the results presented in Figures 1D and E of this article, I would like to inquire why the authors focused solely on detecting the expression of the JEV NS1 protein, rather than examining other non-structural proteins, such as NS2, NS3, NS4, and NS5. Could you please provide an explanation? Additionally, concerning the expression of the GAPDH protein in Figure 1D, why is the expression of GAPDH in the mock group higher than that in the JEV-GFP replicon group? I would appreciate your clarification on this matter.

2. Regarding the analysis presented in Figure 2, which examines the effects of various antiviral drugs on the expression of the NS1 gene in the JEV-GFP replicon, I would like to inquire why methoxsalen, which demonstrates superior inhibition of NS1 gene expression compared to HZ-157, was not selected for subsequent experiments, while HZ-157 was chosen instead. Please provide clarification on this matter. Additionally, concerning the experimental design depicted in Figure 2, why did the authors limit their analysis to the effects of antiviral drugs on NS1 gene expression, rather than investigating other non-structural proteins, such as NS5 or RNA-dependent RNA polymerase? I would appreciate further explanation.

3. Concerning the analysis presented in Figure 3, which examines the effects of JNJ-A07, HZ-1157, NITD-2, quinine, and NITD008 on the expression of NS1 in the JEV-GFP replicon, I would like to inquire how the authors decided to use a concentration of 10 μM for subsequent experiments based on the results shown in Figure 3. Please provide clarification.

4. Regarding the detection experiment of the cysteine residue 302 at position 4 (C4) of JEV NS1, as illustrated in Figure 5, I would like to inquire how this experimental result correlates with the findings related to NJ-A07, HZ-1157, NITD-2, quinine, and NITD008 concerning NS1 gene expression in this article. Additionally, since the article indicates that cysteine residue 302 at position 4 (C4) of EV NS1 is essential for JEV replication, what prompted the authors to conduct further experiments specifically on this residue in JEV NS1? Please provide clarification. Furthermore, regarding the expression of GAPDH protein shown in Figure 5E, why is the GAPDH protein expression in the mock group higher than that in the JEV-GFP replicon group? I would appreciate an explanation.

Author Response

This article presents the results of antiviral drug screening targeting the non-structural proteins of Japanese Encephalitis Virus (JEV) using a non-infectious JEV-GFP replicon; however, the information provided is too brief. While antiviral drugs that affect the expression of the JEV NS1 gene have been thoroughly screened, there has been no further investigation into the expression of other non-structural protein genes. Additionally, the mechanisms by which these antiviral drugs influence JEV NS1 gene expression have not been explored in depth, which is quite unfortunate. Furthermore, there are several questions that the author needs to address, such as:

Comments 1: Regarding the results presented in Figures 1D and E of this article, I would like to inquire why the authors focused solely on detecting the expression of the JEV NS1 protein, rather than examining other non-structural proteins, such as NS2, NS3, NS4, and NS5. Could you please provide an explanation? Additionally, concerning the expression of the GAPDH protein in Figure 1D, why is the expression of GAPDH in the mock group higher than that in the JEV-GFP replicon group? I would appreciate your clarification on this matter.

Response 1: Thank you for pointing this out. We agree with this comment. We selected the JEV antibodies available in a laboratory for detection, hoping to prove that the replicator constructed in this study can replicate normally and express the protein of the JEV replicon. I used the NS1 antibody in this experiment to detect the expression of NS1 protein, so as to reflect that the constructed replicon gene can replicate normally, not to use NS1 as the research target. Based on your opinion, I supplemented the Western blot detection using NS5 antibody. This replicon can also express NS5 protein. However, for the antibodies that could be detected by IFA, we only had the antibody of JEV NS1. Therefore, only the expression of NS1 was detected by IFA. The reason why the GAPDH expression level in the control group was higher than that in the transfected JEV-GFP replicon group in Figure 1D was that I did not unify the protein concentration of the two groups of samples during Western blot detection, resulting in inconsistent GAPDH expression levels detected in the two groups. This study only focused on whether the constructed JEV replicon protein could be expressed normally, and the loading amount was not always accurate, but it did not affect the judgment of the expression of JEV replicon protein. I reunified the protein concentration of the test samples as much as possible and retested Figure 1D.

Comments 2: Regarding the analysis presented in Figure 2, which examines the effects of various antiviral drugs on the expression of the NS1 gene in the JEV-GFP replicon, I would like to inquire why methoxsalen, which demonstrates superior inhibition of NS1 gene expression compared to HZ-157, was not selected for subsequent experiments, while HZ-157 was chosen instead. Please provide clarification on this matter. Additionally, concerning the experimental design depicted in Figure 2, why did the authors limit their analysis to the effects of antiviral drugs on NS1 gene expression, rather than investigating other non-structural proteins, such as NS5 or RNA-dependent RNA polymerase? I would appreciate further explanation.

Response 2: Thank you for pointing this out. We agree with this comment. Although the drug Methoxsalen significantly reduced the relative expression level of the NS1 gene, its toxicity to cells was relatively high. Therefore, it was not selected for subsequent research.

Our laboratory has primers for detecting NS1 qRT-PCR, so this study chose to detect the mRNA expression level of NS1 gene to reflect the replication of JEV-GFP replicon gene, rather than targeting NS1 as the research target. The effect of detecting the expression level of the JEV NS1 gene in this experiment was to illustrate the RNA replication situation of the JEV replicator, thereby reflecting the influence of antiviral on the genomic replication of the JEV replicator. Therefore, only one gene that could be expressed by the JEV replicator was randomly selected for detection, and no other non-structural protein genes were selected. The main reason why other non-structural proteins NS5 and RNA polymerase were not selected is that only the replicators constructed in this study were initially used as tools to screen antiviral drugs for JEV, and the specific mechanism of action of antiviral drugs has not been studied yet. Antiviral drugs generally act on RNA polymerases such as NS3/NS5. This study is only a preliminary screening of drugs at present, and no specific drug target detection is carried out. However, further research on the target of drug action should be carried out in the future. According to your suggestion, we have supplemented the influence of the expression level of the NS5 gene in supplementary Figure 2, and the result is consistent with the expression result of the NS1 gene.

Comments 3: Concerning the analysis presented in Figure 3, which examines the effects of JNJ-A07, HZ-1157, NITD-2, quinine, and NITD008 on the expression of NS1 in the JEV-GFP replicon, I would like to inquire how the authors decided to use a concentration of 10 μM for subsequent experiments based on the results shown in Figure 3. Please provide clarification.

Response 3: Thank you for pointing this out. We agree with this comment. Based on the CC50 results of this study, a higher concentration within the effective concentration range was selected to ensure that the drug could work and had no cytotoxicity. Meanwhile, I also referred to the dosage of the drug in other literatures. Therefore, I chose a concentration of 10 μM in combination with the literature and the results in Figure 2B. At a concentration of 10 μM, there was no cytotoxicity against A549,BHK-21, and 293T cells. Therefore, a concentration of 10 μM was selected for the subsequent experimental study.

  1. Kiemel, Kroell A.H., Denolly S., Haselmann U., Bonfanti J.F., Andres J.I., Ghosh B., Geluykens P., Kaptein S.J.F., Wilken L., Scaturro P., Neyts J., Van Loock M., Goethals O. & Bartenschlager R. Pan-serotype dengue virus inhibitor JNJ-A07 targets NS4A-2K-NS4B interaction with NS2B/NS3 and blocks replication organelle formation. Nat Commun 15, 6080 (2024). https://doi.org:10.1038/s41467-024-50437-3

S.J.F. Kaptein, Goethals O., Kiemel D., Marchand A., Kesteleyn B., Bonfanti J.F., Bardiot D., Stoops B., Jonckers T.H.M., Dallmeier K., Geluykens P., Thys K., Crabbe M., Chatel-Chaix L., Munster M., Querat G., Touret F., de Lamballerie X., Raboisson P., Simmen K., Chaltin P., Bartenschlager R., Van Loock M. & Neyts J. A pan-serotype dengue virus inhibitor targeting the NS3-NS4B interaction. Nature 598, 504-509 (2021). https://doi.org:10.1038/s41586-021-03990-6

  1. Yu, Jing J.F., Tong X.K., He P.L., Li Y.C., Hu Y.H., Tang W. & Zuo J.P. Discovering novel anti-HCV compounds with inhibitory activities toward HCV NS3/4A protease. Acta Pharmacol Sin 35, 1074-1081 (2014). https://doi.org:10.1038/aps.2014.55
  2. Niyomrattanakit, Chen Y.L., Dong H., Yin Z., Qing M., Glickman J.F., Lin K., Mueller D., Voshol H., Lim J.Y., Nilar S., Keller T.H. & Shi P.Y. Inhibition of dengue virus polymerase by blocking of the RNA tunnel. J Virol 84, 5678-5686 (2010). https://doi.org:10.1128/JVI.02451-09

G.Q. Yao, Yu J.C., Lin C., Zhu Y.J., Duan A.N., Li M.F., Yuan J. & Zhang J.C. Design, synthesis, and biological evaluation of novel 2'-methyl-2'-fluoro-6-methyl-7-alkynyl-7-deazapurine nucleoside analogs as anti-Zika virus agents. Eur J Med Chem 234 (2022). https://doi.org:ARTN 114275. 10.1016/j.ejmech.2022.114275

  1. Malakar, Sreelatha L., Dechtawewat T., Noisakran S., Yenchitsomanus P.T., Chu J.J.H. & Limjindaporn T. Drug repurposing of quinine as antiviral against dengue virus infection. Virus Res 255, 171-178 (2018). https://doi.org:10.1016/j.virusres.2018.07.018

Comments 4: Regarding the detection experiment of the cysteine residue 302 at position 4 (C4) of JEV NS1, as illustrated in Figure 5, I would like to inquire how this experimental result correlates with the findings related to NJ-A07, HZ-1157, NITD-2, quinine, and NITD008 concerning NS1 gene expression in this article. Additionally, since the article indicates that cysteine residue 302 at position 4 (C4) of EV NS1 is essential for JEV replication, what prompted the authors to conduct further experiments specifically on this residue in JEV NS1? Please provide clarification. Furthermore, regarding the expression of GAPDH protein shown in Figure 5E, why is the GAPDH protein expression in the mock group higher than that in the JEV-GFP replicon group? I would appreciate an explanation.

Response 4: Thank you for pointing this out. We agree with this comment. 

The detection experiment of cysteine residues (C4) at position 4 of JEV NS1. The result of this experiment is not related to the drug screening of NJ-A07, HZ-1157, NITD-2, quinine and NITD008 in this article. These are two independent experimental parts, indicating that the JEV-GFP replicon platform constructed in this study has two functions, including drug screening and mechanism study of viral gene replication.

Flavivirus replicons have been used as tools to understand the role of viral nonstructural proteins. The cysteine residue at position 4 (C4) of flavivirus NS1, which is involved in the formation of disulfide bonds, plays a role in the stable folding phase of proteins and may be required for JEV RNA replication. The nonstructural proteins are primarily involved in viral RNA replication, and flavivirus NS1 modulates the early events of viral RNA replication. Moreover, another article in our laboratory indicates that the C4 site is the palmitoylation modification site of JEV NS1 and affects JEV replication (this article is currently being written). Therefore, we selected the fourth cysteine residue site of NS1 that is involved in the formation of disulfide bonds and plays a role in the stable folding phase of proteins to examine whether the fourth cysteine residue site of flavivirus NS1 was required for JEV RNA replication using the JEV-GFP replicon as a tool. It is explained in both the results and discussion sections of the article.(Line 308-314) (Line 437-442). The references are as follows:

M.A. Edeling, Diamond M.S. & Fremont D.H. Structural basis of Flavivirus NS1 assembly and antibody recognition. Proc Natl Acad Sci U S A 111, 4285-4290 (2014). https://doi.org:10.1073/pnas.1322036111

  1. Scaturro, Cortese M., Chatel-Chaix L., Fischl W. & Bartenschlager R. Dengue Virus Non-structural Protein 1 Modulates Infectious Particle Production via Interaction with the Structural Proteins. PLoS Pathog 11, e1005277 (2015). https://doi.org:10.1371/journal.ppat.1005277

Y.L. Ci, Liu Z.Y., Zhang N.N., Niu Y.Q., Yang Y., Xu C.M., Yang W., Qin C.F. & Shi L. Zika NS1-induced ER remodeling is essential for viral replication. J Cell Biol 219 (2020). https://doi.org:10.1083/jcb.201903062

W.C. Brown, Akey D.L., Konwerski J.R., Tarrasch J.T., Skiniotis G., Kuhn R.J. & Smith J.L. Extended surface for membrane association in Zika virus NS1 structure. Nat Struct Mol Biol 23, 865-867 (2016). https://doi.org:10.1038/nsmb.3268

This might be due to the fact that the loading protein concentrations of these two groups of samples were not precisely unified. This study only focused on the effect on the replication ability of the JEV-GFP replicon, and the results of this experiment did not affect the judgment of the results.JEV NS5 was tested to reflect the amount of protein produced by JEV-GFP replicon replication, and to reflect the replication ability of JEV-GFP replicon and JEV-GFP-NS1/C4A.

Reviewer 2 Report

Comments and Suggestions for Authors

In this study the authors generate a JEV replicon with a GFP tag and investigate the antiviral activity of five bioactive compounds, replication of the replicon and expression of NS1, as well as the effect and significance of a key residue in NS1 on replication of the replicon.

              A few comments are as follows-

  • Line 35: the genus was renamed Orthoflavivirus in 2023, please update.
  • Methoxsalen also seems to significantly reduce the relative levels of the NS1 gene expression (Fig 2B). Was there a specific reason for not including this compound in the downstream investigations?
  • It might be helpful to add a few lines about the compounds in context of similar studies with reported antiviral compounds tested against JEV, and whether the most promising candidates have any shared chemical characteristics.
  • Choice of words or sentence structure can be clarified in several places, for example-
    • Line 293, calculate or calculated?
    • Line 322, were the cells transfected with both replicons simultaneously?
    • Line 376, the foreign GFP protein?
    • Line 420 and elsewhere,..the NS1 C4 residue that forms a disulfide bond..?
    • Line 396, add ‘respectively’ at the end of the sentence

Author Response

In this study the authors generate a JEV replicon with a GFP tag and investigate the antiviral activity of five bioactive compounds, replication of the replicon and expression of NS1, as well as the effect and significance of a key residue in NS1 on replication of the replicon.

A few comments are as follows-

Comments 1: Line 35: the genus was renamed Orthoflavivirus in 2023, please update.

Response 1: Thank you for pointing this out. We agree with this comment. I have modified "flavivirus" to "Orthoflavivirus"

Comments 2: Methoxsalen also seems to significantly reduce the relative levels of the NS1 gene expression (Fig 2B). Was there a specific reason for not including this compound in the downstream investigations?

Response 2: Thank you for pointing this out. We agree with this comment. Although the drug Methoxsalen significantly reduced the relative expression level of the NS1 gene, its toxicity to cells was relatively high. Therefore, it was not selected for subsequent research.

Comments 3: It might be helpful to add a few lines about the compounds in context of similar studies with reported antiviral compounds tested against JEV, and whether the most promising candidates have any shared chemical characteristics.

Response 3: Thank you for pointing this out. We agree with this comment. I have added this part of the discussion. “At present, there is no effective antiviral drug for JEV. At present, the research drugs are mainly broad-spectrum antiviral drugs, antiviral drugs targeting nucleic acid, and drugs targeting viral replication protease. The compounds screened out in this study can all exert inhibitory effects on different links of flavivirus replication. JEV also belongs to flavivirus, so it may have a similar mechanism of action against JEV. This can be used as the basis for future research on the specific inhibitory mechanism of these drugs against JEV.”

Comments 4: Choice of words or sentence structure can be clarified in several places, for example-

Line 293, calculate or calculated?

Line 322, were the cells transfected with both replicons simultaneously?

Line 376, the foreign GFP protein?

Line 420 and elsewhere,..the NS1 C4 residue that forms a disulfide bond..?

Line 396, add ‘respectively’ at the end of the sentence

Response 4: Thank you for pointing this out. We agree with this comment. 

Line 300-302, I have revised this sentence. “The SI values of JNJ-A07, HZ-1157, NITD-2, Quinine and NITD008 were calculated respectively.”

Line 316-319, I have revised this sentence. “The transfection of JEV-GFP-NS1/C4A replicon and its parental JEV-GFP replicon expressing wild-type NS1 into BHK-21 cells respectively, showed that the green fluorescence was observed in the cells transfected with the JEV-GFP replicon but not in the cells transfected with the JEV-GFP-NS1/C4A replicon.”

Line 384-386, I have revised this sentence. “The analysis of viral NS1 expression and GFP fluorescence indicated that the constructed JEV-GFP replicon can self-replicate and express the nonstructural proteins and the GFP protein.”

Line 420, I have revised these parts.

Line 407-409, I have added ‘respectively’ at the end of the sentence.

Round 2

Reviewer 1 Report

Comments and Suggestions for Authors

The revised version looks good for me.